# An Overview of Factors Affecting the Functional Quality of Common Wheat (*Triticum aestivum* L.)

**DOI:** 10.3390/ijms24087524

**Published:** 2023-04-19

**Authors:** Ewa Filip, Karolina Woronko, Edyta Stępień, Natalia Czarniecka

**Affiliations:** 1Institute of Biology, University of Szczecin, 13 Wąska, 71-415 Szczecin, Poland; 2Institute of Marine and Environmental Sciences, University of Szczecin, Adama Mickiewicza 16, 70-383 Szczecin, Poland

**Keywords:** wheat, GBSS, waxy, PPO, LOX, *Pina-D1*, *Pinb-D1*, environmental factors

## Abstract

Wheat (*Triticum aestivum* L.) is one of the most important crops worldwide, and, as a resilient cereal, it grows in various climatic zones. Due to changing climatic conditions and naturally occurring environmental fluctuations, the priority problem in the cultivation of wheat is to improve the quality of the crop. Biotic and abiotic stressors are known factors leading to the deterioration of wheat grain quality and to crop yield reduction. The current state of knowledge on wheat genetics shows significant progress in the analysis of gluten, starch, and lipid genes responsible for the synthesis of the main nutrients in the endosperm of common wheat grain. By identifying these genes through transcriptomics, proteomics, and metabolomics studies, we influence the creation of high-quality wheat. In this review, previous works were assessed to investigate the significance of genes, puroindolines, starches, lipids, and the impact of environmental factors, as well as their effects on the wheat grain quality.

## 1. Introduction

Due to the enormous global importance of wheat, *Triticum* species have been the subject of intensive research on the origin, taxonomy, and inheritance of their individual traits. The biological enhancement of wheat is mainly related to improving the quality of proteins and enriching them with appropriate amino acids and trace elements, as well as strengthening resistance to unfavorable environmental factors. In recent years, methods describing the intraspecific and interspecies polymorphism of storage proteins, starch, lipids, and DNA have found the greatest application in studies on species of the genus *Triticum* [1,2,3].

The composition and proportions of individual gluten fractions are the main determinants of the rheological properties of dough, which, in turn, indicate the quality of a given wheat. Therefore, most of the parameters of the multi-feature method that allows wheat classification into specific qualitative groups concern quantitative and qualitative gluten assessment. However, there are also other factors that, in addition to proteins, may affect the characteristics of the resulting dough and determine grain suitability for specific purposes. The activity of amylolytic enzymes in wheat grain is one such factor [4,5].

Starch is the most abundant component of the grain endosperm in wheat. It consists of glucose, amylose, and amylopectin polymers. Generally, carbohydrates can be divided into two main types: available and unavailable. The available carbohydrates are those digested and absorbed by humans, which include (non-resistant) starch and soluble sugars. In contrast, unavailable carbohydrates (dietary fiber) are not digested by the endogenous enzymes secreted into the human gastrointestinal tract [6]. Dietary fiber mainly consists of resistant starch (RS), cellulose, and other complex polysaccharides such as arabinoxylans, β-glucans, pectins, arabinogalactans, and lignin [6]. The expression levels of genes encoding enzymes involved in the starch synthesis pathway, as well as genes encoding α-amylase inhibitors, may be important for the baking value [5]. An example could be the gene encoding ADP-glucose pyrophosphorylase, which catalyzes the conversion of glucose-1-phosphate to ADP-glucose, which is subsequently converted to amylose and amylopectin [5,7].

Essential fatty acids (palmitic and linoleic), fat-soluble vitamins, and phytosterols are important components of wheat grain lipids. Based on their solubility under certain extraction conditions, lipids are classified into starch lipids, both free and bound, and non-starch lipids (NSL). Wheat flour lipids are NSLs, which are mainly composed of triglycerides and other nonpolar (NP) lipids such as digalactosyl diglycerides (DGDG) [8]. Generally, NP lipids are mainly present in the free NSL fraction, whereas glyco- and phospholipids are mostly protein-bound and occur in the bound NSL fraction. Flour lipids have a positive effect on dough and loaf volume formation during the bread-baking process, whereas free fatty acids in NP lipids negatively affect loaf volume; glycolipids in polar lipids have a positive effect on this feature. Moreover, loaf volume is influenced by polar and NP lipids. Lipids also affect the volume and softness of steamed bread and the morphology of shortcrust pastry [8].

Climatic conditions, stress factors, and agricultural management are also important in terms of grain quality. The quality parameters of flour and dough may change depending on the harvest year [9]. For example, high temperature negatively affects starch accumulation and, thus, yield. At high temperatures, starch accumulation can be up to 130 times faster, resulting in lower expression levels and activities of genes encoding certain enzymes involved in the sugar synthesis pathway [7]. It has been indicated that plants have various mechanisms regulating and maintaining an appropriate level of macronutrients in seeds and, for example, the effect of elevated temperature on gluten protein composition and proportions may vary among genotypes. Among others, an increase in the gliadin fraction in relation to glutenins and a decrease in flour baking value have been observed [7].

Another aspect is fertilization, which has a significant impact on the development of qualitative characteristics. Providing specific nitrogen doses can positively affect the overall grain protein content and the HMW-GS: LMW-GS ratio. A positive correlation of nitrogen fertilization has been shown for parameters such as sedimentation index, flour water absorption, and dough resistance, although the values of these parameters were mainly related to an individual wheat variety [7,9,10]. On the other hand, sulfur deficiency causes an increase in the ω-gliadin fraction poor in sulfur compared to other proteins [7] or unacceptably low dough elasticity and strength [11]. The application of sulfur in fertilization increases the gluten content and reduces its spreadability. However, the dough gains less resistance to stretching and greater extensibility. 

This review provides the current state of knowledge on the genes of gluten, starch, and lipid complexes responsible for the synthesis of major nutrients in wheat (*Triticum aestivum*) grain endosperm. Allelic and genetic relationships between genes are also discussed, and areas for further research are identified. The review concludes by discussing the abiotic factors responsible for shaping wheat quality traits. 

## 2. Factors Affecting Quality Traits of Common Wheat

For many years, gluten was the basic molecular target of studies on wheat genetics, i.e., a protein complex mainly responsible for the rheological properties of wheat dough. The nature of the proteins included in this polymeric complex has been confirmed in numerous studies, attributing their physicochemical properties to a specific role in bread production [12]. By elastic and cross-linked structure formation, gluten can trap gas bubbles during dough fermentation, which affects the specific and characteristic porous structure of the crumb in wheat bread [10,13]. However, when studying wheat genetics, other wheat grain constituents also need to be considered, as they can have a significant impact on the expression of proteins of the gluten complex as well as of other genes, directly affecting the quality of wheat products. The qualitative wheat characteristics are usually conditioned by the expression of multiple genes, their mutual interactions, as well as their complexity and the inevitable influence of the environment and external factors [14]. The method of post-harvest processing or grain storage has a significant impact on the wheat end product. The factors directly affecting the functionality of wheat quality traits can be divided into two main groups: environmental and genetic factors. By contributing to the production of appropriate physicochemical and biochemical or rheological properties, they influence the formation of a wheat product with favorable baking properties [15]. Subsequently, the obtained breeding effects are subjected to a series of tests aimed at confirming the achievement of the preferred baking characteristics. Kernel hardness is defined as the mechanical force used to crush wheat kernels. A major part of the wheat kernel is the starchy endosperm, which is covered by multiple tissue layers, including the outer pericarp, inner pericarp, testa, the hyaline layer (nucellar layer), and the aleurone cells. The quality of the produced grain is analyzed, among others, by measuring hardness, glassiness, and seed size, as well as grain number and shape [1]. Thus, the wheat seeds have multiple tissue layers, which affect their physical and biochemical properties and are determinants of their qualitative and quantitative properties (Figure 1) [6].

## 3. The Main Groups of Genes Responsible for Wheat Quality Traits

### 3.1. Starch Properties—Waxy (Wx) Genes

A key element of wheat flour is starch, which can make up to three-quarters of its total ingredient pool [16]. It remains an indispensable component in the baking industry due to its functions during bread baking. Wheat starch is responsible for gluten obtaining the appropriate consistency, providing a sugar substrate, and gelatinization, thereby affecting the flexibility and absorbency of the dough for bread making [17]. In relation to grain dry matter, starch accounts for 65–75% of its weight and can also exceed 80% of the endosperm volume [18]. It is in its tissues, or, more precisely, in the amyloplasts, where the synthesis of starch components occurs [19], starch is deposited in separate granules [18]. As a result, after completion of the grain ripening stage, three separate starch fractions are formed: types A, B, and C. The initiation of type A synthesis takes place at the initial stage of wheat grain development, around 4–5 days after the wheat flowering period, and the final amount is reached approximately 7 days later. The smaller type B begins to form much later—about 14 days after flowering, and the development lasts until day 21 after flowering. The third and smallest—type C—is formed during the final stage of grain development [20]. All of the listed types, regardless of the stage of their formation and size, share a common structure. They consist of two key polysaccharides that make up the starch polymeric structure, amylose and amylopectin [2]. In terms of spatial arrangement, amylose has a simpler and less branched structure than amylopectin, which contains both α-1,4-glycosidic and α-1,6-glycosidic bonds. Amylose has only α-1,4-glycosidic bonds, which means that it exhibits a linear chain structure, as opposed to amylopectin, which has an extensive and branched spatial structure.

The overall quality of starch, and thus that of the final wheat product, is influenced by both its granule structure and the quantitative ratio of amylose and amylopectin, which are 32–25% and 68–75%, respectively [5]. The synthesis of both polysaccharides, as mentioned earlier, is performed in the plastids of endosperm tissues, where various enzymes are involved in the mechanism of amylose and amylopectin formation, including several isoforms of starch synthase [2]. Despite the differences between the synthesis pathways of both polysaccharides, ADP-glucose is the substrate initiating the formation in both cases. The formation of amylopectins involves side-branching enzymes (SBEI or SGP-2, SBEIIa, and SBEIIb); debranching enzymes (DBS), which hydrolyze α-1,6-glycosidic bonds; and three classes of starch synthase (SSI or SGP-3, SSII or SGP-1, and SSIII), as shown in Figure 2 [5]. 

In turn, the *GBSSI* gene, which encodes the Waxy enzymatic protein (E.C. 2.4.1.242), plays a key role during amylose synthesis, showing its highly specific activity only in a complex with starch (Figure 2). Dissociation of the complex causes the loss of enzymatic activity by the Waxy protein [5]. In hexaploid wheat, due to its genomic structure, three genes responsible for Waxy (Wx) protein expression can be distinguished: *Wx-A1*, *Wx-B1*, and *Wx-D1*, located on chromosomes 7AS, 4AL (translocation with 7BS), and 7DS, respectively [21]. Hence, three isoforms of these proteins were identified by 2D-PAGE electrophoresis—Wx-A1, Wx-B1, and Wx-D1. In addition, sequencing analysis of all three genes provided a detailed understanding of their structures. Initially, only the coding regions of the *Waxy* genes were considered, and 11 exons and 10 introns were distinguished in their structure [21]. At present, it is known that each of them consists of 12 exons and 11 introns (Figure 3), and the protein mass encoded by them is approximately 60 kDa [5].

The quantitative proportion of amylose in individual wheat cultivars is dictated by the presence or, on the contrary, the lack of Wx proteins involved in its synthesis [22]. Wild-type wheat, in which all three *Waxy* (*Wx*) loci can be found, is called “nonwaxy wheat”. It should therefore come as no surprise that it has the highest amylose content among all wheat varieties [23]. However, any deviation from the wild genotype manifests itself in a change in the level of its synthesis, resulting in wheat with properties completely different from those of the original type. Spontaneous mutations leading to the loss of one or two *Wx* loci result in partial mutants exhibiting an amylose gradient [5,23].

Additionally, numerous analyses of the relationship between the mutation of a specific *Waxy* locus and amylose content allowed for a deeper understanding of the molecular basis of this relationship and the determination of their frequency in response to the geographical distribution of individual wheat cultivars. The *Wx-B1* (*Wx-B1b*) and *Wx-A1* (*Wx-A1b*) null alleles have been identified as occurring most frequently, not only in Japanese native wheat varieties but also in the regions of North America and Europe [24]. The *Wx-A1* (*Wx-A1b*) null allele is highly common in Asia, especially in Korea and Turkey, whereas the *Wx-B1* null allele (*Wx-B1b*) occurs in Australia and India. The only case of the *Wx-D1* null allele (*Wx-D1b*) identification has been recorded in the Chinese wheat variety Bai Huo, which proved its extremely rare occurrence [24]. The presence of each of these *Waxy* allelic variants differently affects the level of synthesized amylose, thereby changing the functional wheat properties to a different extent. Yamamori and Quynh [16], crossing the Japanese wheat variety with the Chinese one, showed that the reduction of the grain amylose content was mainly influenced by the *Wx-B1b* allele and, to a lesser extent, by the *Wx-D1b* allele, whereas the *Wx-A1b* allele exerted the smallest effect. Other researchers reached similar conclusions [25,26,27], namely, that the lack of the Wx-B1 protein had the greatest effect on reducing amylose production compared to other waxy proteins. The effect of each of these mutations can undergo a compensatory process due to the alloploid nature of the common wheat genome. Consequently, the absence of one or two *Waxy* loci may be masked by the activity of the wild-type allele [28]. 

Miura et al. [26], in their study conducted on monosomic wheat lines, noted the possibility of compensating for the effects of the lack of the *Wx-A1b* or *Wx-D1b* alleles through the presence of a protein product of the active *Wx-B1* allele. Hence, it is extremely difficult to find a complete waxy wheat variety, in which amylose is not produced or the process occurs to a negligible degree, in natural conditions, [28,29]. Therefore, it was helpful to crossbreed partial *Waxy* mutants or to intentionally induce mutations, which, in turn, allowed for the creation of a complete *Waxy* wheat variety with a complete set of three null alleles—*Wx-A1b*, *Wx-B1b*, and *Wx-D1b* [2,30]. For this purpose, Japanese bread wheat Kanto 107 (*Wx-A1b*/*Wx-B1b*) and Chinese wheat Bai Huo (*Wx-D1b*) [2] have been most frequently applied. The introduction of the practice of crossing wheat with different waxy allelic variants allowed obtaining several wheat varieties that differ in the presence of individual Wx proteins. 

As mentioned earlier, depending on the area, wheat has found an application in many regional grain products [5]. For this reason, the properties of each of them are subject to different production requirements, and these, in turn, depend mainly on wheat grain components, including starch. The amylose content of wheat flour has a significant impact on the texture, stability, and viscosity of the end wheat products. Its effect on the processed wheat product is mainly found in the production of noodles served in Asia, whose good quality is a consequence of its high swelling volume, sticking capacity, and viscosity peak [5]. Consumers’ requirements for the structure, viscosity, and color of noodles, especially regarding white, salty Udon noodles, are dictated by the preferred wheat varieties, which contain a moderate amount of amylose, i.e., the starch component [16,31,32]. Many researchers have found a beneficial effect of amylose content reduction on the quality of noodles served in Asia. Guo et al. [33] also observed that the addition of waxy flour significantly increased the swelling volume and water absorption capacity of the flour used to make salty Asian noodles. In addition, pasta with the addition of waxy flour is characterized by a slightly lighter color and a much smoother surface than that made from wild wheat varieties. Such a positive effect of the low amylose content on noodle quality has been observed at 20–30% of waxy flour content and 70–80% of wild-type flour [34]. Baik and Lee [35] showed that while a small addition of waxy flour had a positive effect on the properties of Udon noodles, waxy flour applied alone was not suitable for their production. The complete elimination of amylose from flour, or too high a proportion of waxy flour, can result in a loss of the preferred properties of every wheat end product, yielding wheat products that are too sticky, lumpy, or, in the case of bread, with a less crunchy texture. Morita et al. [36] tested flour varieties with different amylose contents and found that those in which the amylose content was too high, or those containing only waxy flour, were not suitable for good-quality bread making. This is because, compared to nonwaxy flour, the structure of the waxy dough is weaker and less stable, and the form and appearance of the bread is less optimal. Hung et al. [37] reached similar conclusions, noting weaker properties of bread obtained from a completely waxy flour. 

In addition to the low loaf volume, the dark brown color and bitter taste were also indicative of its poor quality. On the other hand, as an additive to wild-type flour, waxy flour significantly improved the baking properties, as observed by many researchers [29,36,37]. Due to its high protein and dietary fiber contents, it directly increased the nutritional properties of wheat bread [34,37]. In addition, waxy flour turned out to be highly stable during cooling and more resistant to freezing and thawing, thereby providing an opportunity to improve the characteristics of wheat products stored in freezers and refrigerators, which is aimed at extending the freshness of not only wheat products. On the other hand, the addition of waxy wheat flour, besides increasing the bread’s resistance to extreme conditions during freezing, also provides more effective refreshment after reheating [36]. Peng et al. [34] observed an extremely important role of the addition of waxy flour in the prevention of stale bread formation because its proper addition guaranteed the maximum extension of bread shelf life. Similar conclusions have been reached by Morita et al. [36], who reported that an additive in the form of waxy flour significantly delays the aging of breadcrumbs. In addition, these authors also noted an improvement in the bread loaf volume, its sticky texture, and organoleptic parameters such as improved chewing properties. 

### 3.2. Lipoxygenase Activity—LOX Genes 

Fatty acids, occurring in wheat kernels mainly as polyunsaturated lipids rich in linoleic and linolenic acid, constitute 2.1 to 3.8% of the grain reserve components [38]. In addition to the proteins forming the gluten complex, they have a significant impact on flour baking quality, especially regarding wheat bread. Wheat flour lipids have been classified into two groups: polar lipids, which include glycolipids and phospholipids, and non-polar lipids, which include mono-, di-, and triglycerides with free fatty acids [8]. The group of polar lipids that are complex with proteins and starch plays a particularly important role here, having a beneficial effect on bread volume [39,40]. Chung et al. [8] observed that the consequence of a higher amount of the polar lipid fraction in wheat flour was a significant increase in the volume of the bread loaf. 

The group of glycolipids, constituting an integral component of the gluten complex, seems to be extremely important in this case. Glycolipids, due to hydrogen bonds and hydrophobic interactions, form connections between key gluten proteins, taking part in shaping their spatial structure. In practice, such aggregation results in an improved ability of the wheat dough to retain gas bubbles and, consequently, has a positive effect on the breadcrumb structure [40]. On the other hand, the non-polar lipid group contributes to the baking properties of wheat, thereby reducing the quality of the baked bread. 

Lipid oxidation is a completely natural phenomenon accompanying the stages of wheat processing such as grain processing or storage. As a reaction dependent on many factors, it may positively or negatively affect the quality of wheat and wheat products. A negative consequence of oxidation is a decrease in the nutritional value of cereals and a loss of color from cereal products, which, in some cases, may be regarded as a positive feature of cereal products [41]. On the other hand, its beneficial effects include an improvement in dough-mixing tolerance [42,43] or, at the micro-level, an increase in the number of disulfide bonds (-SS) [42].

Lipid oxidation can occur via an enzymatic pathway or without the participation of reaction catalysts. The main enzyme involved in the oxidation of cereal lipids is lipoxygenase (EC 1.13.11.12)—an oxidoreductase which is a non-heme iron-dependent protein. It catalyzes the oxidation of polyunsaturated fatty acids, especially linoleic and linolenic acids, accompanied by the release of peroxides and hydroperoxides [41,44,45,46]. In wheat, the highest concentration of lipoxygenase (LOX) is recorded in the seed coat and embryo [47], whereas its content in the endosperm itself is relatively low [42]. In addition, there is little information on genes encoding LOX in polyploid plants such as common wheat (*Triticum aestivum* L.) [44]. However, so far, it has been possible to localize two *LOX* loci—*Lpx-1* and *Lpx-2*—on chromosome groups 4 and 5, respectively. In the following years, the *LOX* genes were intensively studied. Garbus et al. [48] identified four *Lpx* sequences, including the *Lpx* pseudogene in genome A (*Lpx-A1_like)*, the *Lpx-B1.1a* and *Lpx-B1.2* genes in genome B, and *Lpx-D1* in genome D. Feng et al. [44] identified and cloned cDNA sequences of the *LOX* genes. The result of their research was the acquisition of the full-length nucleotide sequences of the *TaLOX1* and *TaLOX2* genes and their assignment to the short arm of chromosome 4D and the long arm of chromosome 5D, respectively. Subsequently, information on the *TaLOX3* gene was provided by Feng et al. [49]. They cloned its complete sequence and mapped it to chromosome 4A. The cloning of the *TaLox-B1* gene located on chromosome 4BS, supported by experimental validation, was conducted by Geng et al. [50], who obtained the full-length sequence of its genomic DNA. On the same chromosome, Zhang et al. [51] cloned two new genes, *TaLox-B2* and *TaLox-B3*.

Due to their unique nature, lipids present in wheat kernels are one of the main research subjects related to wheat baking quality. They are characterized by a high proportion of unsaturated fatty acids, whose presence enables redox reactions with the thiol groups of wheat proteins [42]. Initially, only the coupling of intermediate lipid oxidation products with the oxidative changes occurring during dough mixing was suspected [52]. Subsequent studies, however, provided more information on the relationship between lipid oxidation and the role of intermediates formed during oxidation on the baking properties of wheat dough. On the other hand, the very nature of lipoxygenase has become a key substrate of these reactions. The ability of LOX to whiten and decolorize wheat dough is one of the earliest known effects of this enzyme [53]. Since lutein, which is one of the main carotenoids of wheat seeds responsible for their yellow color, has an antioxidant effect, it is partially destroyed under the influence of oxygen alone. This mechanism is additionally induced by high temperatures, light, and hydroperoxides formed in the reaction of lipid oxidation in the presence of lipoxygenase [54]. Research on its impact on the quality of the obtained bread was carried out, among others, by Bahal et al. [42], who noted significant differences between bread obtained from flour supplemented with LOX and those without supplementation. The presence of lipoxygenase resulted in a larger loaf volume, a more delicate texture, and, above all, a lighter crumb, which seemed to be advantageous features in the baking industry, especially in white bread production [55]. Leenhardt et al. [54] focused on the causes of gradual color loss by wheat products and noted a decrease in lutein content in response to LOX activity stimulated by the presence of water during the wheat dough mixing step. As a result of lipid oxidation and the formation of LOX intermediates, the wheat baking product was deprived of some of the stored pigments, resulting in its whitening. 

In addition to the already described consequences of lipoxygenase action on wheat dough, its effects have also been observed, inter alia, in the improved mixing tolerance. From a technological point of view, this is the potential of the dough to counteract excessive mixing and prevent its decomposition after reaching optimum development. Faubion and Hoseney [53] proved the role of LOX in a significant increase in wheat dough mixing tolerance by adding an actively enzymatic soybean flour rich in lipoxygenase. The improvement of this technological factor is possible due to the double role of LOX. On the one hand, it is a radical activator, and, on the other hand, its activity causes the formation of free fatty acid radicals [53]. Its detailed action can be explained by the example of fumaric acid, whose effect on wheat flour was investigated by Sidhu et al. [56]. It is a double-bond compound that binds to the thiol groups of gluten proteins formed after breaking the disulfide groups during mixing. This weakens the structure of the wheat dough, thereby reducing its tolerance to mixing. These authors also noted that the active form of fumaric acid increases the number of free radicals during mixing, which additionally affects the degradation of the dough structure and the weakening of the mixing tolerance. On the other hand, lipoxygenase, because of its properties, reverses the harmful effects of this compound, creating free fatty acid radicals that can compete for compounds with double bonds in their structure. This prevents the binding of fumaric acid to the thiol groups of proteins, thereby restoring and strengthening mixing tolerance [43,53]. 

Soy flour, rich in LOX isoenzymes, has a positive effect on the formation of disulfide bonds when added to wheat flour. Wheat dough supplemented in this manner was characterized by a lower number of free thiol groups and more SS (disulfide) bonds. Thus, the action of LOX initiates the formation of new links between SH groups due to the oxidation of thiol groups by oxidation reaction products [42]. The elimination of thiol groups in favor of new disulfide bonds strengthens the gluten structure but also reduces the elasticity of the wheat dough. Therefore, high-quality flour should be optimized in terms of the LOX content because the extensibility and elasticity of the dough, which mainly affects its strength, determines the quality of the obtained bread. Such an optimum would not cause the formation of an excessive number of SS bonds, and thus would not reduce the quality of wheat dough, and it would have a positive effect on the baking properties of common wheat [45]. 

### 3.3. Polyphenol Oxidase Activity—PPOs—Polyphenol Oxidase Genes

Polyphenol oxidases are the main factors responsible for the undesirable darkening of the final wheat products, such as chapatti bread, steamed bread, or Asian noodles, especially Udon or Cantonese types [57,58,59]. This time-dependent color change caused by the activity of PPOs can be explained by the oxygen-dependent mechanism of action of the enzymes in a given group. The PPOs exhibit two enzymatic activities—monophenol mono-oxygenase and tyrosinase or cresolase activity (EC 1.14.18.1), in addition to diphenol oxidase or catecholase activity (EC 1.10.3.2) [60,61]. Hence, they are capable of catalyzing two consecutive reactions. In the first one, they cause o-monophenol hydrolysis, and, as a result, o-diphenols are formed, which are subsequently oxidized in the second reaction to o-quinones [61]. The resulting highly reactive quinone products can interact with other nucleophiles and can modify and cross-link other phenolic products, interact with amino acids and thiol groups of proteins, or undergo self-polymerization. This results in the formation of polymeric brown melanin-like complexes, manifested by the discoloration of wheat products [61,62,63,64]. The characteristic structure of PPOs is responsible for their involvement in redox reactions. They belong to metalloproteins, containing two conserved domains, CuA and CuB, each of which has three histidine residues. The structured enzyme center is its reactive site, coordinating copper ions and interacting with molecular oxygen and phenolic substrates [62,65]. Such substrates include, among others, catechin, pyrocatechol, L-DOPA, methyl catechin, chlorogenic acid, and L-tyrosine. A multitude of substrates results in the presence of many isoenzymes in polyphenol oxygenases [66].

Molecularly, polyphenol oxygenases belong to the multigene family encoded in the cell nucleus [67]. They were classified into paralogs that arose as a result of duplication and subsequent mutations [64,68]; this classification was described for the first time by Jukanti et al. [62], who showed the presence of two separate phylogenetic groups, each consisting of three members. While genes of one group are expressed in developing wheat grains, genes belonging to the other group are expressed in tissues other than seeds [69]. Many attempts have been made to map them in the common wheat genome. As a result of these actions, the genes controlling PPO activity were found mainly on homologous group 2 chromosomes [70,71,72]. However, several authors also noted their locations on chromosomes of groups 3, 5, 6, and 7 [70]. However, the major locus controlling polyphenol oxidase activity in hexaploid wheat is located on the long arm of chromosome 2A [71,73] and on the long arm of chromosome 2D [73,74]. One of the paralogical *Ppo-1* families, represented by the *Ppo-A1* and *Ppo-D1* genes, has been localized on these chromosomes. Both their position and function have been intensively studied [74,75,76,77]. The allelic variants of both *Ppo-A1* and *Ppo-D1* are closely related to the activity of PPOs in wheat grains [59,74,78,79,80,81]. He et al. [74] developed two complementary STS markers, PPO16 and PPO29, facilitating the successful identification of two alleles located on chromosome 2DL, *Ppo-D1a* (713 bp) and *Ppo-D1b* (490 bp), associated with low and high PPO activity, respectively. However, in terms of the impact on the activity of polyphenol oxygenases, a greater effect was recorded in the case of the *Ppo-A1* variant [79,80]. The presence of a second paralogical *Ppo-2* family was identified by Beecher and Skinner [82]. The described *Ppo-A2, Ppo-B2*, and *Ppo-D2* genes were also localized on the chromosomes of homologous group 2, on the long arm of chromosome 2A, the long arm of chromosome 2B, and the long arm of chromosome 2D, respectively [75].

Polyphenol oxidases are highly ubiquitous enzymes in plant tissues and are mainly located in plastids, whereas their potential substrates are present in the vacuoles [83]. Hence, their direct contact and darkening reactions occur during mechanical damage of the tissues during wheat grain processing, including flour milling [84]. The presence of PPOs in wheat flour is a consequence of the trace amounts of bran present. The activity of polyphenol oxygenases was localized in the bran, which is a by-product of the milling of purified wheat grain [68]. Hence, one way to reduce the amounts of PPOs in flour is their removal just before grain milling [85]. In addition to the undesirable darkening of wheat end products, many other functions related to the activity of PPOs in plant organisms have been described so far, most notably, those associated with the defense response of plants. 

There is substantial evidence that genes encoding PPOs can be induced by various stressors such as pathogens, pests, wounds, or even herbivores, resulting in tissue-specific expression of proteins from a particular enzyme group [63,86]. However, despite the evident role of PPOs in the defense against these stressors, more detailed information regarding the protection mechanism itself is still lacking [86]. Due to their activity, PPOs can occur in both latent and enzymatically active forms. Activation of the latent form of the enzyme can occur under the influence of chemical compounds such as acids, detergents, or alcohols, which can lead to the removal of the enzyme inhibitor, changes in its conformation, and, consequently, PPO activation [61,87]. Okot-Kotber et al. [87] reported the superiority of ionic detergents over their non-ionic forms in stimulating the latent form of polyphenol oxidase. However, it should be remembered that the action of PPOs is mainly and closely associated with the genotype [88]. Polyphenol oxidase activity of durum wheat (*Triticum durum* Desf.) is significantly lower than in common wheat (*Triticum aestivum* L.) [89,90]. Therefore, variability in PPO activity can be observed among wheat cultivars and different locations of their cultivation [88]. The continuously changing climate and environment of a given growing region play a role in determining PPO activity. Beta et al. [84] also proved the influence of the wheat cultivation environment on the dynamics of PPO activity. Such results suggest that the activity of PPOs is the product of numerous factors, caused by genotypic traits, environmental conditions, and the technology of wheat processing and cultivation. 

### 3.4. Puroindoline Genes

Wheat owes its wide application in the grain industry to several unique features, making it a versatile product. Although several specific wheat parameters are guaranteed to obtain appropriate baking and quality properties, the texture and hardness of its grains seem to be the features that ensure wheat universality as a cereal product. It is the multitude of varieties that ensures its wide application in the global grain industry market [91]. Hexaploid wheat was classified as soft or hard based on the grain structure, determined by measuring its resistance to milling (Table A1, Table A2 and Table A3) [92]. The choice of an individual variety and, consequently, the texture of its grain defines the final wheat product, directly influencing the grain milling process, flour baking, and the overall rheological quality of wheat dough [93]. Milling of soft grains is relatively easy, resulting in fine-grain flour with a higher content of undamaged starch granules, a lower protein content, and a lower water absorption capacity [94]. 

A greater problem in milling the endosperm is encountered in hard-grain wheat, whose structure requires more energy during milling [95]. Detailed research on energy requirements was carried out by Szabó et al. [96] showing a significant relationship between high energy consumption during grain milling, water absorption, and hard wheat grain. Coarse flour obtained from such wheat is characterized by a high density of physically damaged starch granules, the amount of which increases with increasing grain hardness [94]. The level of such damage is important in the selection of different flour types for the production of specific wheat products. Due to its properties, hard-grain wheat is of the greatest importance for the baking and milling industry (flour production) [97]. The flour obtained from it is mainly characterized by high water absorption, which is caused by both the high content of degraded starch granules and the high protein proportion in its composition [94,98]. In addition, due to the suitable baking properties of the flour extracted from hard-grain wheat, bread obtained from it is characterized by a large volume of loaf crumb [96]. The high level of starch granule degradation, resulting in a high water absorption capacity, means that coarse flour is mainly applied in the baking of bread and wheat products with yeast leaven. Soft-grain wheat flour is used in the production of cakes, pastries, and other confectionery products [99,100]. 

Grain hardness is mainly controlled by the expression of proteins of the *Ha* locus (*Hardness*), located on the short arm of chromosome 5D. These loci have also been identified on the remaining homologous chromosomes of group 5 in hexaploid wheat, but there are some differences between them [101]. Genes related to grain texture determination among the *Ha* loci on chromosomes 5AS, 5BS, and 5DS are located only on the last one of the listed chromosomes. These are the *Pina-D1* and *Pinb-D1* genes, which encode two types of puroindolines—PINA and PINB, respectively—and the *Gsp-D1* gene, responsible for the expression of the GSP-1 protein (grain softness protein 1) [102]. However, there is no evidence of any involvement of GSP-1 in grain texture determination [95,103]. Two alleles that determine individual traits are distinguished in terms of the genotype related to the hardness of wheat grains. The dominant *Ha* allele is associated with soft-wheat grains, whereas the *ha* allele is related to the hard variety [91,104]. The grain hardness phenotype is a consequence of the presence or absence of friabilin, a protein composed of two lipid-binding proteins—puroindoline a (PINA) and puroindoline b (PINB) [105,106]. Hence, when the friabilin content is low, it is a hard wheat, whereas high levels result in grains with a soft texture [103,107]. On the other hand, the presence of only PINA and PINB proteins will not produce a phenotypic effect of soft grain texture without the formation of complex bonds with starch [92]. This is possible due to the presence of polar lipid residues on its surface, forming bonds with the puroindolines [108]. Moreover, the high affinity of PINA and PINB for polar lipid residues on the starch surface is ensured by the tryptophan-rich domains present in their structure [109,110]. It is the binding of puroindolines to lipids on the starch surface that affects the adhesion of the starch-protein matrix, representing the essence of the formation of a specific wheat grain texture and dictating the type and quality of the final wheat product [102]. Many researchers have linked the presence of mutations in the *Pina-D1a* and *Pinb-D1a* genes with wheat grain hardness [109,111,112,113]. While soft wheat carries the wild form of both puroindoline-coding genes (*Pina-D1a/Pinb-D1a*), the hard variety has a mutation in at least one of them [92]. Alleles of the wild type—*Pina-D1a* and *Pinb-D1a*—determine soft grains. Mutation in one or both Pin genes increases grain hardness. In the case of the *Pin* genes, alleles carrying different mutations have been detected (Table A1 and Table A2) [114]. The first mutation in the *Pina* gene, reported by Giroux and Morris [111], was its complete deletion, resulting in the *Pina-D1b* wheat genotype (null allele). This is one of the most common puroindoline mutations. Another mutation is the substitution in the *Pinb* gene, replacing glycine with serine at position 46—*Pinb-D1b* [111]. In both cases, the consequence is a completely different degree of grain hardness. The wheat of the *Pina D1b* genotype shows much greater grain hardness than that carrying the *Pinb-D1b* allele, which was observed by Chen et al. [115], as per Martin et al. [116], and Chen et al. [101]. In consequence, the *Pinb-D1b* mutation results in a greater flour gain and bread with a larger loaf volume than in the presence of the *Pina-D1b* mutation [117]. Under natural conditions, spontaneous mutations in both loci, or the deletion of the entire *Ha* locus, in common wheat are extremely rare [105]. Such a genotype, with a double null mutation of both genes encoding puroindolines a and b, was noted by Ikeda et al. [118] and Tanaka et al. [119]. Due to the lack of these alleles, the tested cultivars did not have any PIN proteins, resulting in an unusually high hardness of their grains. Such a texture does not have a positive effect on wheat baking properties, and, hence, the double null mutants usually do not find their application in the baking industry. The research of Huertas-García et al. [120] showed remarkable variability in the puroindoline genes. One of the four mutations found in *Pinb-D1* had not been previously described, was considered novel, and was named *Pinb-D1ak*. This mutation is in the tenth cysteine of the protein backbone, which is a highly conserved region. So far, 28 *Pina* alleles and 33 alleles of the *Pinb* gene have been described in Table A1 and Table A2, and many unique alleles are still detected, such as *Pinb-D1g*, which has been found among landraces from Poland [114,121]. The large family of 44 Pin genes identified in common wheat significantly impacts its growth and development. Thus, all plant scientists have been focused on studies of this family of genes because it significantly influences stress tolerance and improves the quality of the crop [90,114,122].

The measurement of grain hardness is possible thanks to direct studies of its structure or indirect analyses of its composition. Due to a large number of detected allelic variants of puroindolines, wheat grains can be divided according to single kernel characterization system (SKCS) hardness in a continuous unit (namely hardness index, HI) ranging from 0 to 100, distinguishing soft (HI < 40) and hard kernels (HI > 60) [90]. The SKCS (single kernel characterization system) is used to directly assess grain hardness by measuring the force required to crush subsequent grains subject to a given analysis. It takes into account their weight, diameter, and degree of hydration [96,123]. We also distinguish measurement methods such as PSI (particle size index) and NIR (near-infrared reflectance) [124,125]. The first one determines the hardness of the grain by its grinding and sieving through a sieve with a specific aperture. The NIR method indirectly enables the assessment of particle size by measuring absorbance using an interferometer [91,126]. On the other hand, the SDS-PAGE technique is a more available method for friabilin content measurements, whose analysis suggests the type of grain hardness. In addition, PCR (polymerase chain reaction), with the use of specific markers to analyze the presence of specific puroindoline alleles, is applied in several grain texture measurement techniques [123]. 

## 4. The Impact of Environmental Factors on Quality Traits of Common Wheat

Environmental conditions have a twofold effect on wheat grain development. Before flowering, they affect germination, ear development, inflorescence formation, and photosynthesis, which contribute to the number of wheat grains. On the other hand, after the flowering period, environmental conditions have a direct influence on seed size and its composition [7]. The composition and interactions between individual components of the grain play a key role in the development of the rheological properties of wheat dough. The content and accumulation of protein in wheat kernels are two of the main parameters indicating the good quality of wheat itself and of the baking flour obtained from it [127]. Protein composition, the molecular weight distribution of individual fractions, and their solubility are considered to be variables that influence the formation of the qualitative wheat properties. Therefore, most environmental research is still focused on them. External conditions can cause changes in the secondary, tertiary, and quaternary structure, thereby affecting the structures and functions of individual proteins [128]. Additionally, stress caused by the influence of environmental factors, acting at the grain-filling stage, may result in negative changes in the quality of the obtained wheat. Stressors can lead to reduced wheat grain filling, accelerated cell death rate, and earlier crop maturity. As a result, changes are recorded in the proportions of individual protein fractions and starch grains [4]. Hence, the most important environmental factors having a direct impact on the growth and development of wheat include water access and temperature conducive to its proper growth [4,129]. Particularly noteworthy is the latter environmental factor. 

Against the background of a warming climate, cereal cultivation will inevitably be impacted [130]. This, in turn, reduces wheat baking qualities, negatively affecting the baking industry. Hence, agriculture, with the help of modern molecular biology techniques, is forced to constantly observe the consequences of environmental factors to counteract the loss and deterioration of yield quality characteristics. Thus, despite the constantly changing climate, most of the analyses allow to predict the effects and implement possible procedures aimed at improving wheat quality. However, despite continuous research into the genetic and environmental interactions of wheat, there is still too little information about the nature of non-protein components and their correlation under constantly changing environmental conditions.

### 4.1. Influence of Temperature

The optimum temperature specified to achieve maximum wheat yields falls within the range of 15–20 °C. In this temperature range, the grain-filling period is maximally extended, resulting in the highest degree of starch accumulation in the seeds, leading to the production of good-quality wheat [7]. Any deviation from the optimal temperature for wheat cultivation results in changes in its physiology, with the risk of a decrease in the quality and baking properties of wheat. Even a short period of increased temperature to which its developing grains are subjected may result in their reduced quality. As a consequence, the period of grain development and water uptake, kernel dry matter accumulation (including proteins and starch), and the stage of their dehydration are reduced [131]. In addition to the stress factor itself, i.e., high temperature, in this case, the plant developmental stage, during which it will be subjected to a given stressor, is also important. The greatest influence of high temperature on grain quality is observed during the grain-filling stage and between the stem elongation and flowering stages, resulting in reduced grain size and a limited grain number per yield [132]. Wardlaw [133] and other researchers [130] noted a clear decrease in grain weight, which was indisputably associated with the effect of high temperature during its development. The reduction in grain weight caused by the shortening of the grain-filling stage additionally entails another consequence for the grain industry, namely, a significant decrease in wheat yields [134,135]. In such reduced wheat grains, differences at the molecular level are found, which are a consequence of changes occurring under the influence of high temperature. Both the content and size of wheat grain components, such as starch or individual fractions of the gluten complex, are modified.

The increase in the flour protein content, induced by thermal stress, has been reported widely [132,136,137] and has been unambiguously confirmed. The degree of accumulation of this key component of the wheat grain is recorded at a different rate at each stage of plant development [14]. The strategic phase of flour protein formation is the grain-filling stage, during which it receives all the necessary chemical components. Temperatures that are too high, together with frequent droughts, can significantly shorten this developmental stage, which directly affects the contents of individual protein fractions in wheat grains [138]. During the grain-filling stage, the accumulation of gliadins occurs much earlier than that of glutenins, and a shortening of this phase also results in an increase in the concentration of the gliadin fraction and a decrease in the glutenin fraction [4,130,139]. Temperatures that are too high, together with frequent droughts, can significantly shorten this developmental stage, which directly affects the contents of individual protein fractions in wheat grains [138]. Daniel and Triboi [134] reported a higher percentage of both protein and gliadin fractions following an increase in temperature. In addition, as reported by Altenbach et al. [131], the accumulation of individual fractions of the gluten complex during increased temperatures ceased much earlier than usual under physiological conditions of wheat cultivation. 

Thermal stress additionally affects the starch content in the endosperm of the kernels. As noted by Hurkman et al. [18] and other researchers [130,135,136,140], high temperature influencing the grain-filling stage led to both starch quantity reduction and changes in the distribution of its individual granules in the mature grain. There is a decrease in the level of type B starch granules along with an accompanying increase in the accumulation of type A starch granules [18,136,140]. Limited duration of the starch-accumulation stage alone largely results in its reduced content, as proved by Altenbach et al. [131]. On the other hand, a decrease in the starch amount may be related to the thermal inactivation of enzymes that play a strategic role in the biosynthesis of this polysaccharide component [132,135].

### 4.2. Influences of Drought and Water Availability

Water scarcity can affect every stage of wheat development, from seed germination to harvest. However, susceptibility to water stress is not equal for every stage of plant development. While some developmental stages can cope with water scarcity, others respond with extreme sensitivity [141]. Particularly susceptible to water deficit are the division and development of cells, whose reduction directly restricts the growth of individual plant organs. In addition, limited water availability reduces the intensity of net photosynthesis and transpiration capacity and induces quantitative changes in endogenous hormones [142]. The adverse effects of drought may therefore result in obtaining wheat of a changed quality and in significantly reduced yields. The crops of wheat grown in continental, semi-arid, and Mediterranean climates are particularly vulnerable to the negative effects of water deficit, but even transitional periods of drought in humid and maritime climates can result in crop losses. Drought is a natural phenomenon, but the constant intensification of climate change in favor of its warming [143] poses constant challenges to grain processing. Hence, a better understanding of the impact of stress associated with the phenomenon of water deficit is crucial in preventing both qualitative and quantitative losses of wheat crops.

Several genes have been described that are activated at the transcriptional level, with cis- and trans-acting factors involved in the expression of dehydration-responsive genes. The dehydration-responsive element binding (DREB) family of transcription factors (TFs) represents one of the major players involved in abiotic (dehydration, cold, high saline) stress responses. The dehydration-responsive element-binding proteins (DREBs) are important transcription factors that interact with a DRE/CRT (C-repeat) sequence and are involved in the responses to multiple abiotic stresses in plants. Latini et al. recently analyzed the expression of the target gene (*Wdnh13*) of the *TdDRF1* at consecutive plant growth stages from different durum wheat genotypes evaluated in two different field environments. Their research found analogies with the transcript profiles, and the results of the qRT-PCR highlighted differences in molecular patterns, thus suggesting a genotype dependency of the *TdDRF1* gene expression in response to the induced stresses. Furthermore, these authors used a statistical association between the expression of *TdDRF1* transcripts and agronomic traits, stating that one of the genotypes was found to combine molecular and agronomic characteristics [144].

Many authors have shown the importance of water deficit on wheat properties [4,144,145,146,147]. However, how drought affects wheat and its yield is closely related to the period of stressor application as well as to its duration and intensity [148]. Exposing the plant to a stress factor after flowering has the greatest impact on the properties of wheat. The availability of water is largely responsible for both the yield and quality characteristics of wheat [149]. Stress caused by the absence of water during the early stages of kernel development or the seed-filling stage significantly reduces the size and final dry matter of kernels [147], whereas the lack of water before and during the wheat flowering period contributes to the reduction of its ears and grains per ear, also affecting grain yields [150,151]. The period when cultivation is subject to drought conditions may consequently lead to yield reduction, with a frequently altered grain composition [152].

Gooding et al. [153] showed that water deficiency is the most important factor in shortening the grain-filling stage, especially when wheat experienced drought effects between days 1 and 14 after flowering. Similar conclusions were drawn earlier by Shah et al. [154], who observed a clear reduction in the grain dry weight accumulation period, as well as decreased cell division, as a result of subjecting wheat to water stress. The consequence of shortening the time of this important stage of cereal development is the undeniable reduction in the dry weight of wheat kernels. Numerous studies have demonstrated the loss of dry weight as a result of water stress during the early stages of grain filling [133,150,152,154,155]. The growth and development of the grain depend on the presence of carbon supplied from the current assimilation, the remobilization of assimilates obtained by the plant before flowering from the stems and other organs, as well as the re-translocation of assimilates stored temporarily in the stem after flowering. A water deficit after the flowering period effectively reduces carbon assimilation from stems, and the availability of current assimilates is important for grain-filling stage [147,156]. Thus, drought may limit assimilation of supply to grains, resulting in a reduced final weight [154]. Such poor grains with reduced dry matter content consequently exhibit a poorer milling capacity and lose baking quality [148]. 

The reduction in the duration of the grain-filling stage related to weight loss also affects the size of the obtained grains. Khan et al. [147] showed a significant decrease in their size as a result of limited water availability during the early stages of grain filling. Similar conclusions were also reached by Wang et al. [157] and Schmidt et al. [148]. Shortening the grain-filling stage, which leads to a reduction in kernel weight and size, ultimately leads to quantitative yield losses [158,159]. Reduced wheat crops due to water stress applied during the early stages of grain filling were recorded, among others, by Khan et al. [147], Balla et al. [4], and Potopová et al. [160]. On the other hand, the water deficit applied at each reproductive stage of wheat may lead to a reduction in grain yield, which has already been proven by various authors [133,151,161,162,163].

Although drought exerts negative effects, resulting in significant crop losses, it can also have a positive influence on certain wheat grain components. Many authors have shown a significant increase in the protein content of wheat flour as a result of stress related to water deficit [146,152,164,165]. Similar conclusions were also reached by Ozturk and Aydin [150] and Fan et al. [166], who, in addition to the elevated protein concentration, also observed an increased wet and dry gluten content. Changes in the composition of wheat flour proteins due to water stress can be explained by the increased nitrogen accumulation index and the decreased value of the carbohydrate accumulation index [167]. The greatest impact of stress on nitrogen increments is observed at the final stages of grain filling, between 1 and 14 days after flowering [153]. Increasing the total flour protein content is not always accompanied by a uniform increment of its individual fractions. As Saint Pierre et al. [146] observed, the increase in the percentage of monomeric proteins during a reduced water supply was greater than that of polymeric proteins. On the other hand, Zhang et al. [168] confirmed that the amount of the gliadin fraction increased during the application of drought stress, whereas the glutenin fraction remained unchanged. Moreover, Panozzo et al. [169] recorded a significant increase in the size distribution of polymeric proteins under water deficit conditions, whereas the gliadin-to-glutenin ratio remained stable. Such different protein profiles of wheat tested under reduced water supply not only result from the influence of environmental factors but mainly from the genotype of a given wheat variety and its possible interactions with the environment [170,171]. Panozzo and Eagles [172] found that the glutenin content in wheat flour was more strongly correlated with the wheat genotype than with environmental factors, whereas the gliadin content was highly dependent on the environment but less on the individual wheat variety. 

Changes at the level of protein fractions caused by a reduction in water supply lead to alterations in the baking properties of the obtained wheat. Li et al. [164] observed that changes in the rheological properties of wheat dough could concern an increase in its strength and endurance, associated with a decrease in extensibility and a reduced loaf volume, yielding a poor-quality wheat end product. On the other hand, Zhou et al. [173] suggested that drought conditions may improve the rheological properties of wheat dough; they tend to improve the gluten properties, including an increase in the contents of glutenin macropolymers (GMP), resulting in greater loaf volume and improved bread quality. Along with the evaluation of the protein content, the SDS value also allows for a fairly accurate prediction of the wheat baking potential [150,174]. Water deficit conditions have a positive effect on the SDS, causing its increase, as observed by Ozturk and Aydin [150], Houshmand et al. [174], and Fan et al. [166]. Hence, on the one hand, drought can lead to the production of good-quality wheat, and, on the other hand, it can result in unfavorable baking properties. Differences resulting from the quality of the obtained wheat products may result from the use of wheat varieties with varying sensitivity to water stress. As suggested by Li et al. [164], wheat varieties with at least moderate gluten strength can avoid the potential negative effects of drought. 

Changes in the level of starch synthesis also occur during stress associated with water scarcity. Many authors have observed a relationship between the effects of water stress and changes in the quality of the produced starch [144,149,154,175]. Singh et al. [176], studying the effect of water deficit after the flowering period, noted significant changes in the size of individual starch granules in wheat grains. While the size of type B and C granules decreased in response to stress, the size of type A granules increased. Moreover, the authors clearly indicated that the distribution of size alterations was closely related to the genotype of a given wheat variety subjected to water deficit conditions. Hence, not all of the studied varieties were characterized by the same proportion of starch granules as a consequence of the stressor. On the other hand, Zhang et al. [149] observed that water stress also had a significant effect on the volume and surface area of starch granules. In addition, the latter researchers emphasized that the wheat genotype played an important role in defining both parameters. 

While some wheat varieties showed a reduction in granule volume due to water deficit, in others, the volume increased or remained unchanged. The same was true for their surfaces. The reasons for changes at the level of starch synthesis can be found in the activity of key enzymes involved in its biosynthesis. Several of them, including granule-bound starch synthase (GBSSI/Waxy) or soluble starch synthase, lose their activity under water stress. Hence, the divergent effect of water deficit, which, on the one hand, leads to a decrease in starch biosynthesis and thus grain yield and, on the other hand, to an increase in protein content and the possibility of obtaining good-quality baked goods, begins to appear comprehensible [173].

The described effects of water deficiency on wheat cultivation and quality are not entirely clear. The nature of the interactions between the genotype of a given wheat variety and the environmental factors appears to be of key importance in this case, and it seems to be related to genetic control due to the varying degrees of stability of specific wheat varieties under stressful conditions [171]. Hence, testing the water deficit tolerance of various wheat varieties offers new perspectives in breeding programs. Wheat with new genotypes that can sufficiently cope with water scarcity conditions is usually produced by crossbreeding potentially stress-resistant wheat [177]. Many authors have studied specific wheat genotypes in the context of their sensitivity to drought conditions and identified both susceptible and resistant varieties [141,170,178]. As noted by Guóth et al. [179], the stability of a given genotype to drought should be inferred based on the overall plant’s response to stress, as the analysis of only the physiological parameters of vegetative plant parts leads to fragmentary conclusions.

### 4.3. Influence of Nitrogen

While soil mineralization is insufficient to meet the nitrogen requirements of plants, various means of supplementation of this element in the form of fertilizers are usually applied [180]. However, there are many reasons for limiting their use in plant breeding practices. Currently, many countries are imposing strict restrictions on the widespread use of nitrogen fertilizers, which is related to the growing awareness concerning environmental protection and the economic situation of the country. This is because the implementation of nitrogen fertilization, in addition to beneficial effects on plant cultivation and breeding, also generates huge costs and high environmental pollution through groundwater contamination [181]. 

However, the important role of nitrogen cannot be denied, as it is crucial in supplying cereal grains with a complete set of protein components. Nitrogen fertilization is considered one of the most important abiotic factors responsible for the increased distribution of wheat flour proteins and, thus, also individual components of the gluten complex [182]. Johansson et al. [183] and other researchers [184,185] agree that nitrogen supplementation significantly increased the contents of both gliadins and glutenins. However, as noted by Triboi et al. [186] and Saint Pierre et al. [146], the content of gliadins increased significantly faster than that of glutenins with an elevated nitrogen supply. The consequence of this effect is an increase in the gliadin-to-glutenin ratio and the HMW-GS-to-LMW-GS coefficient [184,187]. The consequence of a higher ratio between both gluten fractions is an increased bread loaf volume [188] and a reduced strength of the gluten complex [185]. In addition to interfering with the protein composition of wheat grain, nitrogen fertilization also affects the yield, which is associated with an increased weight of wheat kernels [189,190,191]. Moreover, nitrogen application also has a beneficial effect on the hardness of wheat grains, resulting in an increased percentage gain of wheat flour [187]. Hence, a significant reduction in nitrogen-containing fertilizers may have serious consequences not only for agriculture but also for the baking industry. The result of such restrictions is usually a significant reduction in the content of wheat storage proteins. 

In addition, Johansson et al. [183] noted that a decrease in nitrogen supply did not affect the formation of new shoots but led to the reduction of the existing ones. However, it should be noted that in addition to the nitrogen amount, its source, plant supplementation period, and fertilization technique are also important [183,184]. Wuest and Casman [190] observed that the application of nitrogen fertilizer before sowing had little effect on the subsequent nitrogen uptake by wheat after the flowering period. In turn, fertilization during the flowering stage significantly enhanced this process. The late application of nitrogen fertilizer can therefore ensure an adequate amount of nitrogen during the grain filling stage, which, in turn, will lead to an increase in protein content [192]. Additionally, as reported by Johansson et al. [185], early nitrogen application results in the production of gluten with greater strength compared to late supplementation.

All reports on the effective use of nitrogen in wheat cultivation suggest that its rational use in the grain industry guarantees a high quality of wheat products. As nitrogen deficiency is one of the key factors limiting the efficiency of wheat cultivation [193], implementing the most efficient use of nitrogen in agriculture, taking into account the reduction of adverse effects on the natural environment, guarantees a good wheat product and low economic and environmental losses.

### 4.4. Influence of Sulfur

The primary role of sulfur during plant developmental processes has been known for over two centuries. However, until the 1980s, the use of additional sources of this element in plant cultivation was not practiced because its deficiency was not detected at that time. Air pollution caused by thriving industrial plants, resulting in acid rain, provided the soil with sufficient sulfur amounts needed for the proper growth and development of crops. Tightening measures introduced in the late 1980s regarding sulfur dioxide (SO_2_) emissions, as well as the use of more concentrated phosphate fertilizers with lower sulfur content, eventually led to significant reductions in the amount of sulfur in the soil. Hence, despite the low crop demand for sulfur, its deficiency has become one of the main factors limiting wheat yields in many regions of the world [194,195,196,197,198]. Consequently, it became inevitable to introduce additional supplementation in the form of sulfur fertilizers. 

Plants have a natural ability to store sulfur during short-term deficiencies. However, they are not able to cope with long periods of sulfur deficit [199]. Sulfur shortage, both in wheat and other higher plants, first manifests itself in the form of chlorosis of young leaves. Mature leaves, on the other hand, may still remain green, indicating their relatively constant sulfur content [200]. Its amount in vegetative tissues, in which it is present in the form of sulfates, can reach up to 50% of the total sulfur content. Mature wheat grain contains only 1–5% sulfur, of which 80–95% is deposited in amino acids, cysteine, and methionine, and the rest is present in the form of sulfates [198,201]. Wieser et al. [202] studied the effect of the sulfur deficit on the presence of key amino acids in wheat grain. They recorded a significant decrease in the amount of both cysteine and methionine. Cysteine, participating in the formation of disulfide bonds between adjacent chains of polypeptides of the gluten complex, plays a key role in shaping its spatial structure [202]. Hence, such a decrease in sulfuric amino acids consequently reduces the rheological parameters of wheat dough, negatively affecting its viscoelastic properties [203]. This was confirmed by Järvan et al. [204], who observed an increase in the contents of cysteine and methionine, as well as threonine and lysine, in response to an increase in sulfur supply. Such an unfavorable effect of sulfur deficiency on the amount of key amino acids in wheat grain simultaneously increases the levels of other amino acid components. 

As reported by Granvogl et al. [205], the asparagine (Asn) content of flour is significantly reduced along with an increase in sulfur supply, whereas the deficit of this element leads to higher quantities of Asn. Wrigley et al. [206] also observed an increase in the proportions of arginine (Arg) and aspartic acid (Asp) in wheat grains obtained from crops with reduced sulfur supplementation. Such changes in grain amino acid composition, caused by sulfur deficiency, affect the synthesis of individual wheat proteins [197]. This especially concerns the decrease in the amounts of sulfur amino acids. Numerous studies have shown a reduction in the quantities of sulfur-rich protein fractions and an increase in sulfur-poor fractions following a reduction in the amount of this component during the development of wheat grains [192,207,208]. Wieser et al. [197] performed a detailed analysis of the changes in the proportions of individual protein types in response to reduced sulfur availability. Despite the small effect of sulfur deficiency on the total gluten content, it turned out that its individual fractions underwent a significant quantitative change. There was a drastic increase in ω-gliadins and a moderate one in HMW-GS glutenins, whereas the levels of γ-gliadins and LMW-GS glutenins were significantly reduced. On the other hand, the quantity of α-gliadin decreased only slightly. As a consequence, the ratio of gliadins to glutenins was markedly increased [209].

Considering the key role of proteins of the gluten complex in forming the appropriate wheat baking properties, changes in the composition of its individual fractions also entail modifications in wheat dough properties. Sulfur deficiency usually means decreased dough extensibility, which causes a reduction in the amount of the LMW-GS fraction [209]. In parallel, the strength of the wheat dough increases, and, as a result, the texture of the bread itself deteriorates, and the loaf volume decreases [195,210]. The reduced baking volume is, in fact, not caused by an excessively soft dough but rather by excessive hardness. As a consequence, fermentation gases are not adequately retained by the stiff structure of the wheat dough, and thus loaves with a lower porosity and volume are formed [211]. Li et al. [212] observed similar changes in the rheological wheat parameters, additionally reporting an influence of sulfur deficit on dough strength.

In addition to the described effect of a decreased sulfur supply on the composition of quantitative protein fractions, a lower sulfur amount can also change the grain structure. This is consistent with the results of Ruiter et al. [208], who found a clear increase in wheat grain hardness in response to a reduced sulfur supply. Flour obtained from the milling of such grains usually yields a dough with a texture that is too hard and too inelastic, which, in turn, leads to a reduction in the quality of the final wheat products [204].

## 5. Conclusions 

Summarizing the currently available literature, we note that the most significant parameters directly affecting wheat quality traits and, consequently, the quality of wheat flour include both genetic and non-genetic factors. Among them, we distinguish the main lipid, carbohydrate, and protein groups and genes encoding them, e.g., HMW-GS, LMW-GS, Waxy, PPOs, PINA, and PINB, as well as environmental factors responsible for shaping the quality characteristics of wheat grain. Temperature, water availability, and fertilizers applied by farmers are the most important abiotic factors affecting wheat development and grains, ultimately conditioning its weight and endosperm composition. Insufficient wheat hydration will result in obtaining flour from grains with increased protein content, mainly due to high nitrogen accumulation and low deposition of carbohydrates. Excessive moisture will, in turn, adversely affect the polymerization of wheat grain storage proteins. Publications on wheat rheology have indicated that exposing wheat to temperatures above 35 °C for several days results in obtaining flour with poor baking qualities. 

Another important issue concerns the intensive application of nitrogen fertilizers, leading to increased protein content in wheat flour, often accompanied by a deficiency of sulfur, the lack of which has a negative impact on disulfide bond formation. As a consequence, insufficient cross-linking of the gluten complex occurs, ultimately leading to wheat flour with poor baking quality. Hence, knowledge of the exact polymorphism of proteins, lipids, carbohydrates, and the genetic basis of their occurrence is essential for the continuous improvement of the technological characteristics of common wheat. For many years, numerous studies have been carried out on the structures of the gluten, lipid, and carbohydrate complexes that play key roles in proper wheat development. Factors affecting the quality of wheat are being continuously investigated, and the role of non-protein components, such as lipids or carbohydrates, indicating their interaction with gluten proteins, is being identified or excluded. We assign them a role in the aggregation of gluten proteins and the formation of the cross-linked gluten structure, with a direct influence on the formation of good-quality wheat baking goods. In summary, ongoing research on genetic and environmental interactions in wheat strives to complement the knowledge of protein and non-protein components and their correlation with constantly changing environmental conditions. The review of numerous research results presented in this article will contribute to an increasingly comprehensive understanding of the mechanisms governing the functional quality of common wheat.

## Figures and Tables

**Figure 1 ijms-24-07524-f001:**
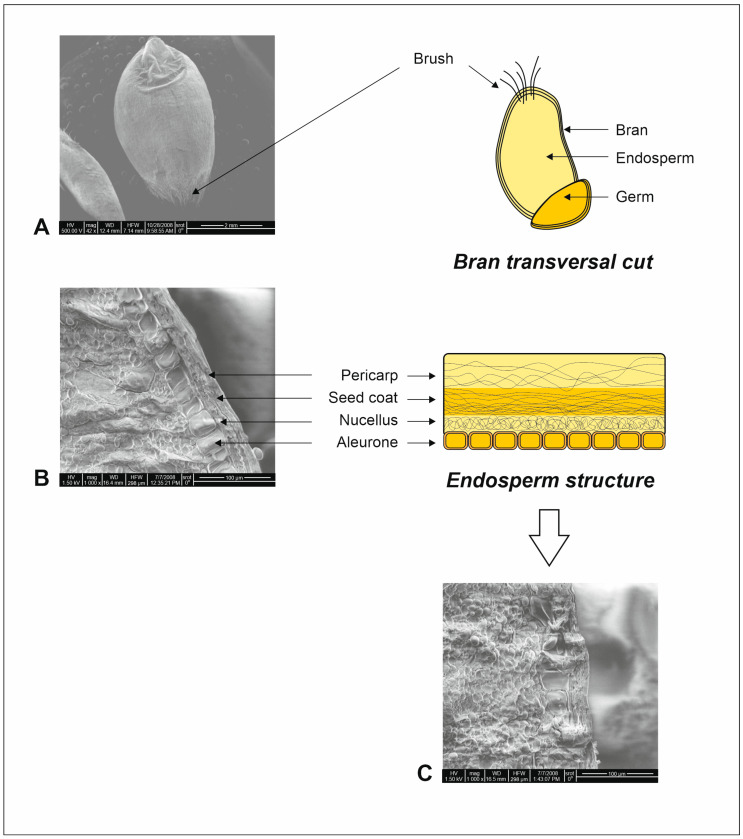
The wheat kernel shows the three major, distinct parts: germ, endosperm, and bran (pericarp), along with the bran transversal cut cross and endosperm structures. The wheat grain contains 2–3% germ (lipids, plants sterols, antioxidants, vitamins, minerals, enzymes); 13–17% bran (non-starch polysaccharides, metabolic proteins, minerals, vitamins, triglycerides) and 80–85% mealy endosperm (protein, starch, fibers). (**A**–**C**): wheat kernel samples observed in scanning electron microscopy (SEM), (**A**): HV: 500 V; mag: 42×; WD: 12.4 mm; HFW 7.14 mm; (**B**,**C**): HV: 1.5 kV; mag: 1000×; WD: 16.4–16.5 mm; HFW 298 µm showing mature wheat kernels.

**Figure 2 ijms-24-07524-f002:**
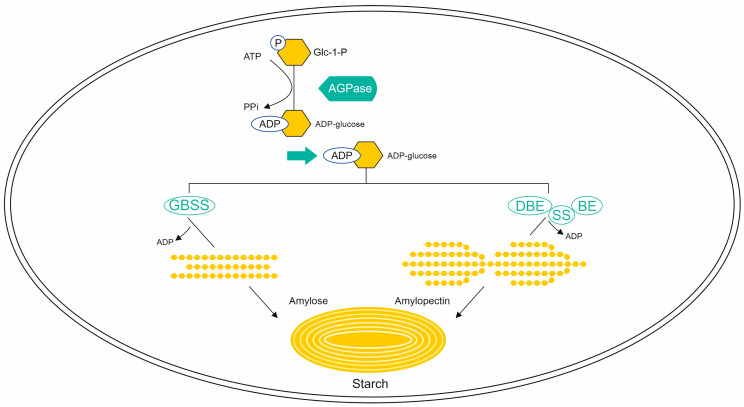
Diagram summarizing the pathway of the formation of the wheat starch granule. In amyloplast, the amylose is synthesized by granule-bound starch synthase (GBSS), while amylopectin biosynthesis requires three more coordinated enzymes of soluble starch synthase (SS), starch-branching enzyme (BE), and starch-debranching enzyme (DBE).

**Figure 3 ijms-24-07524-f003:**
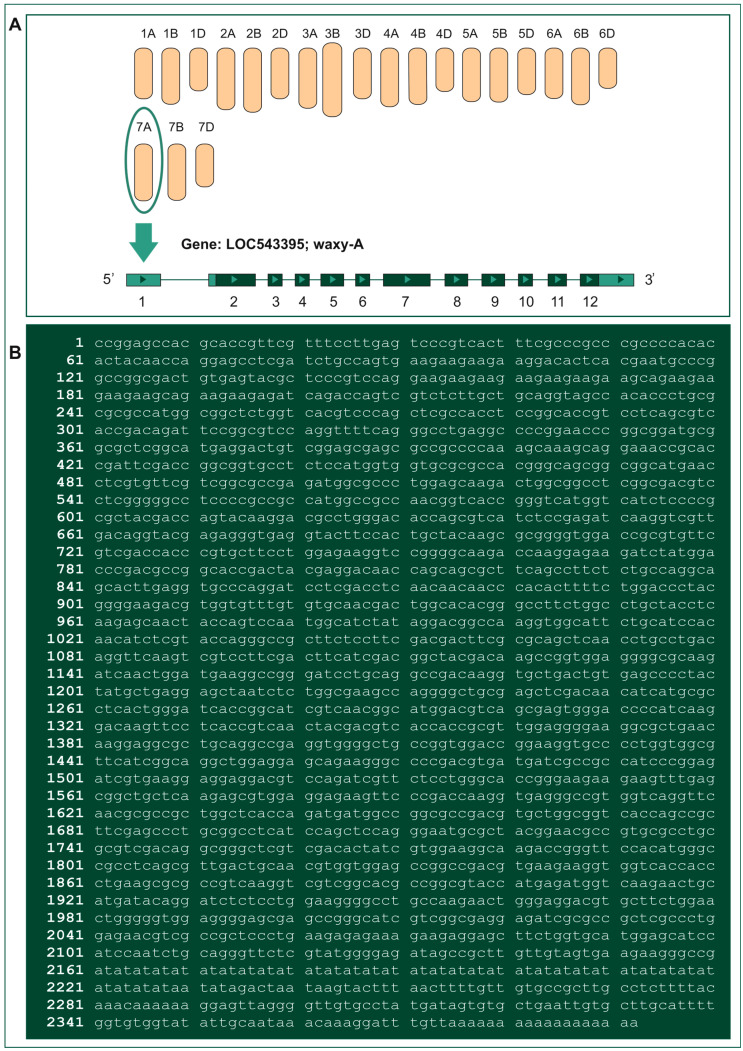
Diagrammatic representation (**A**): of the waxy gene structure, composed of 12 exons and 11 introns Triticum aestivum granule-bound starch synthase 1, (**B**): mRNA; nuclear gene, according to NCBI Reference Sequence: NM_001405816.1.

## Data Availability

The data that support the findings of this study are available from the corresponding author, ewa.filip@usz.edu.pl, upon reasonable request.

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
