# Peer review of "An Overview of Factors Affecting the Functional Quality of Common Wheat (Triticum aestivum L.)"

_ijms, 2023, doi:10.3390/ijms24087524_

Round 1

Reviewer 1 Report

The manuscript is in line with the journal aims and issues.

The English language is optimal, a quick revision is suggested.

Aims of paper:  to increase the knowledge concerning the environmental factors (biotic and abiotic) affecting wheat grain quality and yields performance and focus on the genes controlling the quality traits. 

The paper is well written with a good schematic idea focused on wheat quality genes and factors influencing quality.

All issues are reported in detail on each group of molecules (genes, proteins, sugars, lipids, enzymes) involved in the final phenotype of the complex quality traits. 

The cited literature is abundant and correct, update at 2021, but there are not any citation on 2022. Thus, I suggest to find and add some newer citations or substitute the oldest ones, if they are not irreplaceable. Furthermore, I suggest to add the following papers:

1)    Conxita Royo, Jose Miguel Soriano, Rubén Rufo, Carlos Guzmán, Are the agronomic performance and grain quality characteristics of bread wheat Mediterranean landraces related to the climate prevalent in their area of origin? Journal of Cereal Science, Volume 105, 2022, 103478, ISSN 0733-5210, https://doi.org/10.1016/j.jcs.2022.103478.

2)    Latini, A.; Cantale, C.; Thiyagarajan, K.; Ammar, K.; Galeffi, P. Expression Analysis of the TdDRF1 Gene in Field-Grown Durum Wheat under Full and Reduced Irrigation. Genes 2022,13,555. https://doi.org/ 10.3390/genes13030555 

3)    Dreisigacker S, Xiao Y, Sehgal D, Guzman C, He Z, Xia X, et al. (2020) SNP markers for low molecular glutenin subunits (LMW-GSs) at the Glu-A3 and Glu-B3 loci in bread wheat. PLoS ONE 15(5): e0233056. https://doi.org/10.1371/journal. pone.0233056 

4)    Battenfield SD, Sheridan JL, Silva LDCE, Miclaus KJ, Dreisigacker S, Wolfinger RD, et al. (2018) Breeding-assisted genomics: Applying meta-GWAS for milling and baking quality in CIMMYT wheat breeding program. PLoS ONE 13 (11): e0204757. https://doi.org/10.1371/journal. pone.0204757 

5)    Latini, A.; Sperandei, M.; Cantale, C.; Arcangeli, C.; Ammar, K.; Galeffi, P. Variability and expression profile of the DRF1 gene in four cultivars of durum wheat and one triticale under moderate water stress conditions. Planta 2013, 237, 967–978 

Author Response

Manuscript ID: ijms-2285701

Dear Reviewer,

Thank you for reviewing a draft of manuscript no “ An overview of factors affecting the functional quality of common wheat (Triticum aestivum L.) ”. We have incorporated most of the suggestions made by the reviews.

In response to comments on the literature, we corrected it by inserting newer editions of 2022. In addition, we added a paragraph regarding the expression of dehydration-responsive genes -Lines 669-679. English corrections have also been made.

All changes to the manuscript are visible with change tracking

Best regards,

Ewa Filip (Corresponding author)

University of Szczecin, Institute of Biology, WÄ…ska 13, 71-415 Szczecin, Poland; [email protected]

Reviewer 2 Report

The work by Filip et al. summarized the significance of genes such as those for grain hardness, starch, and lipids and the impact of environmental factors and their influences on wheat grain quality. The manuscript is largely well written and covered many aspects about wheat grain quality. My major concern is that there is too little new literatures in this review which may indicate insufficient progress in this field. A rough account of papers published after 2016 is 24, ~10% of the total papers cited. Therefore, this reviewer wonders whether it is a right time to write such a review since the subject has been reviewed multiple times in various journals.

Minor concerns:

Some language issues should be checked carefully. Examples are:

1.        Line 18, “influeces” should be “influences”?

2.        Line115, “which are determinants” should be “which are determinants for”?

Author Response

Manuscript ID: ijms-2285701

Dear Reviewer,

Thank you for reviewing a draft of manuscript no “ An overview of factors affecting the functional quality of common wheat (Triticum aestivum L.) ”. We have incorporated most of the suggestions made by the reviews.

In writing this manuscript, the authors wanted to draw attention to factors affecting the functional quality of common wheat (Triticum aestivum L.). We are constantly wondering what else can affect the quality of common wheat. Using the available literature, we drew the reader's attention to wheat; GBSS; waxy; Ppo; Lox; Pina-D1; Pinb-D1; environmental factors. In response to comments on the literature, we corrected it by inserting newer editions.

English corrections have also been made.

All changes to the manuscript are visible with change tracking

Best regards,

Ewa Filip (Corresponding author)

University of Szczecin, Institute of Biology, WÄ…ska 13, 71-415 Szczecin, Poland; [email protected]